# Autophagy in High-Fat Diet and Streptozotocin-Induced Metabolic Cardiomyopathy: Mechanisms and Therapeutic Implications

**DOI:** 10.3390/ijms26041668

**Published:** 2025-02-15

**Authors:** Rong Zhou, Zutong Zhang, Xinjie Li, Qinchun Duan, Yuanlin Miao, Tingting Zhang, Mofei Wang, Jiali Li, Wei Zhang, Liyang Wang, Odell D. Jones, Mengmeng Xu, Yingli Liu, Xuehong Xu

**Affiliations:** 1Laboratory of Cell Biology, Genetics and Developmental Biology, Shaanxi Normal University College of Life Sciences, Xi’an 710062, China; zhourong1993@snnu.edu.cn (R.Z.); zhangzutong@snnu.edu.cn (Z.Z.); lixinjie@snnu.edu.cn (X.L.); duanqinchun@snnu.edu.cn (Q.D.); miaoyuanlin2022@snnu.edu.cn (Y.M.); zhangtingting2022@snnu.edu.cn (T.Z.); wangmofei2022@snnu.edu.cn (M.W.); li-jl83@hp-snnu.edu.cn (J.L.); zhangwei1991@snnu.edu (W.Z.); wly1826@snnu.edu.cn (L.W.); liu-yl7@hp-snnu.edu.cn (Y.L.); 2University Laboratory Animal Resources (ULAR), University of Pennsylvania School of Medicine, Philadelphia, PA 19144, USA; jonesod@exchange.upenn.edu; 3Department of Pediatrics, Columbia University, New York, NY 10032, USA

**Keywords:** metabolic cardiomyopathy, autophagy, high-fat diet (HFD), streptozotocin (STZ), ferritinophagy, ER-phagy

## Abstract

Metabolic cardiomyopathy, encompassing diabetic and obese cardiomyopathy, is an escalating global health concern, driven by the rising prevalence of metabolic disorders such as insulin resistance, type 1 and type 2 diabetes, and obesity. These conditions induce structural and functional alterations in the heart, including left ventricular dysfunction, fibrosis, and ultimately heart failure, particularly in the presence of coronary artery disease or hypertension. Autophagy, a critical cellular process for maintaining cardiac homeostasis, is frequently disrupted in metabolic cardiomyopathy. This review explores the role of autophagy in the pathogenesis of high-fat diet (HFD) and streptozotocin (STZ)-induced metabolic cardiomyopathy, focusing on non-selective and selective autophagy pathways, including mitophagy, ER-phagy, and ferritinophagy. Key proteins and genes such as PINK1, Parkin, ULK1, AMPK, mTOR, ATG7, ATG5, Beclin-1, and miR-34a are central to the regulation of autophagy in metabolic cardiomyopathy. Dysregulated autophagic flux impairs mitochondrial function, promotes oxidative stress, and drives fibrosis in the heart. Additionally, selective autophagy processes such as lipophagy, regulated by PNPLA8, and ferritinophagy, modulated by NCOA4, play pivotal roles in lipid metabolism and iron homeostasis. Emerging therapeutic strategies targeting autophagy, including plant extracts (e.g., curcumin, dihydromyricetin), endogenous compounds (e.g., sirtuin 3, LC3), and lipid/glucose-lowering drugs, offer promising avenues for mitigating the effects of metabolic cardiomyopathy. Despite recent advances, the precise mechanisms underlying autophagy in this context remain poorly understood. A deeper understanding of autophagy’s regulatory networks, particularly involving these critical genes and proteins, may lead to novel therapeutic approaches for treating metabolic cardiomyopathy.

## 1. Introduction

Over the past two decades, the incidence of heart failure has risen significantly, mainly driven by the increasing prevalence of chronic metabolic diseases such as insulin resistance, diabetes, and obesity. These conditions are often associated with excessive caloric intake and lifestyle factors that promote pathological metabolic regulation [1,2]. Metabolic cardiomyopathy, a chronic metabolic disorder, is characterized by structural and functional changes in the heart, including interstitial fibrosis, and occurs in the occurrence of coronary artery disease or hypertension [2]. Diabetic cardiomyopathy and obese cardiomyopathy both fall under the category of metabolic cardiomyopathies, underscoring their shared metabolic origins. Patients with diabetes often exhibit left ventricular systolic dysfunction, which can progress to heart failure [3,4,5]. A meta-analysis by Erqou et al. (2022) demonstrated that increased insulin resistance is associated with a higher risk of heart failure, independent of traditional cardiovascular risk factors [6]. In 2021, Ren et al. further elaborated on the features of obesity cardiomyopathy, noting that long-term obesity is closely linked to cardiac remodeling, which is characterized by left ventricular hypertrophy, fibrosis, and diastolic dysfunction, ultimately leading to heart failure [7]. Type 1 diabetes mellitus (T1DM) and type 2 diabetes mellitus (T2DM) can be induced in C57BL/6J (C57) mice through dietary interventions and different doses of streptozotocin (STZ) [8]. HFD is commonly used to model obesity and T2DM, and are associated with metabolic disorders (Appendix A). Additionally, obesity always affects negatively the development of T2DM; therefore, the progression of T2DM is typically accompanied by obesity. The development of metabolic cardiomyopathies and subsequent heart failure occurs independently of coronary artery disease, hypertension, and vascular heart disease [3,4,7].

Metabolic cardiomyopathy is often accompanied by the abnormal regulation of autophagic homeostasis, a process essential for cellular maintenance physiologically and survival pathologically [9,10]. Autophagy activation contributes to the degradation of damaged proteins and organelles, including collagens, thereby mitigating cardiac fibrosis in metabolic cardiomyopathy [11]. Autophagy can be broadly classified into two types based on the specificity of its targets: non-selective autophagy and selective autophagy. Non-selective autophagy is typically activated in response to nutrient deprivation [12], whereas selective autophagy specifically targets particular cellular contents, often mediated by ubiquitination and selective autophagy receptors [13,14]. In selective autophagy, isolated membrane assembling around specific targets surrounds these targets, and basically constructs the formation of autophagosomes, which facilitates their degradation [13]. Depending on the types of cellular contents being degraded, selective autophagy encompasses various subtypes as well, including mitophagy, ER-phagy, ferritinophagy, lysophagy, aggrephagy, xenophagy, pexophagy, ribophagy, glycophagy, lipophagy, and fluidophagy [15].

Mitophagy plays a crucial role in maintaining mitochondrial homeostasis, and its abnormal regulation is closely linked to metabolic diseases. The ubiquitination-dependent pathway is one of the most extensively investigated mechanisms of mitophagy [16,17]. ER-phagy, another form of selective autophagy, is critical for maintaining the homeostasis of the endoplasmic reticulum (ER) [13]. Recent studies have highlighted the positive regulatory role of Sequestosome 1 (SQSTM1/p62) in ER-phagy, wherein it binds to the transmembrane E3 ligase TRIM13 (tripartite motif containing 13) to promote ER-phagy under conditions of ER stress [18]. Additionally, the ubiquitination of ER-resident proteins mediated by Ufm1-specific ligase 1 (UFL1) ligase plays a pivotal role in regulating ER-phagy [19]. CDK5 regulatory subunit-associated protein 3 (C53), an ER-phagy receptor, is recruited by UFL1 ligase and binds to autophagy-related protein 8 (Atg8), facilitating the delivery of isolation membranes to the ER and promoting ER-phagy [20]. Ferritinophagy, which is vital for maintaining iron homeostasis, involves the release of iron from ferritin stores through autophagy, with nuclear receptor coactivator 4 (NCOA4) serving as a selective cargo receptor [21]. NCOA4-mediated ferritinophagy leads to increased intracellular iron levels, promoting ferroptosis by accumulating iron within cells [22].

Abnormal regulation of autophagy is a hallmark of metabolic diseases, contributing significantly to the development and progression of metabolic cardiomyopathy. The auto-phagic response in diabetic cardiomyopathy varies between T1DM and T2DM [23]. Despite extensive publications identifying molecular mechanisms associated with metabolic cardiomyopathy—such as systemic inflammation [2], fatty acid oxidation [24], disturbances in intracellular calcium homeostasis [25], oxidative stress [26], abnormal autophagy [26], and myocardial fibrosis [27]—the precise role of autophagy in metabolic cardiomyopathy remains incompletely understood. Recent advancements in the field of autophagy in metabolic cardiomyopathy have begun to shed light on the underlying mechanisms. This review will focus on the regulatory pathways of non-selective and selective autophagy in HFD- and STZ-induced metabolic cardiomyopathy, while also exploring emerging therapeutic agents that target autophagy for the treatment of metabolic cardiomyopathy.

## 2. Regulation of Autophagy in Different Metabolic Cardiomyopathy Models

Recent advancements have greatly expanded our understanding of autophagy in metabolic cardiomyopathy. The Zucker diabetic fatty (ZDF) rat model has been extensively used to explore the effects of obesity and diabetes. In this model, basal cardiac autophagy is impaired, as evidenced by a downregulation in the LC3II (lipidated form of LC3)/I ratio, while p62, a selective autophagy receptor, is significantly upregulated [28]. Similarly, in the leptin-deficient ob/ob mouse model, non-selective autophagy is disrupted due to impaired autophagosome formation, as indicated by a failure in LC3 lipidation. In contrast, selective autophagy, as evidenced by increased levels of p62, Atg7, and Rab9 (Ras-related protein Rab-9A), compensates for this deficiency [29,30].

Growing evidence has reinforced the critical role of autophagy in metabolic cardiomyopathy, particularly in the context of obesity and T2DM. Since fasting activates non-selective autophagy, it is unsurprising that overnutrition leads to autophagic inhibition. Several independent studies have shown that chronic HFD consumption significantly decreases autophagic activity in the heart [31,32,33,34,35,36,37,38,39,40]. Specifically, the levels of non-selective autophagy, as indicated by reduced expression of Atg5, Atg7, Atg12, Beclin-1, and the LC3II/LC3I ratio, are significantly downregulated after prolonged HFD feeding in wild-type mouse hearts [31,32,33,34,35,36,37,38,39,40]. Recent studies have also shown that autophagic flux in the hearts of WT mice, as assessed by the LC3II level, initially increases but then decreases in response to HFD, peaking at six weeks and returning to baseline at two months [41].

Autophagic responses in diabetic cardiomyopathy exhibit distinct patterns between T1DM and T2DM [23]. Several studies have demonstrated that insulin inhibits autophagy [42,43]. T1DM mice, which have lower insulin levels than T2DM mice, exhibit controversial findings regarding autophagy in STZ-induced T1DM metabolic cardiomyopathy. Some studies suggest that excessive autophagy in STZ-induced insulin-deficient mice contributes to myocardial injury [44,45]. Evidence from both in vivo STZ-induced diabetic mice and in vitro glucolipotoxicity animal models has highlighted that excessive autophagy in T1DM is a critical risk factor for cardiac dysfunction [46]. However, other studies propose that autophagy is defective in T1DM cardiomyopathy [47,48,49,50,51,52,53]. In STZ-induced T1D mice and cardiomyocytes treated with high glucose and palmitic acid, cardiac non-selective autophagy is inhibited, contributing to myocardial injury [49,50,51,52,53,54].

Recent evidence further supports the knowledge that selective autophagy is also disrupted in metabolic cardiomyopathy. Huang et al. recently reported a significant decrease in the protein levels of Pink1 (PTEN-induced putative kinase protein 1), Park2 (RBR E3 ubiquitin protein ligase gene 2), Atg9, Mfn1 (Mitofusin1), and member RAS oncogene family, Rab7, in mitochondrial fractions of mouse hearts following 20 weeks of HFD treatment [32]. Other studies have shown that PINK/Parkin (Parkin RBR E3 ubiquitin–protein ligase)-dependent mitophagy is impaired under these conditions [24,32], while the alternative Ulk (Unc-51-like kinase 1)/Rab9 mitophagy pathway is activated after prolonged HFD exposure [55,56]. Additionally, ferritinophagy is significantly upregulated in the mouse heart following hyperlipidemia challenge [57]. Recent studies have also indicated that ER-phagy is enhanced by hyperglycemic stimulation in vitro [58].

Taken together, these findings illustrate the dysfunctional autophagy observed in metabolic cardiomyopathy, as summarized in Table 1. Both non-selective and selective autophagy are defective in genetic and HFD/STZ-induced models of metabolic cardiomyopathy. Furthermore, abnormal selective autophagy emerges as an additional risk factor contributing to cardiac dysfunction in these models.

## 3. Non-Selective Autophagy in HFD/STZ-Induced Metabolic Cardiomyopathy

Non-selective autophagy is typically activated in response to nutrient deficiencies, as we discussed above. During non-selective autophagy, an isolation membrane is formed from the ER [13]. AMP-activated protein kinase (AMPK), considered as a critical energy sensor, is regulated by the AMP/ATP ratio and promotes autophagy through phosphorylation of ULK1 and Beclin-1, as well as the inactivation of the mammalian target of the rapamycin complex 1 (mTORC1) [87,88]. The activation of the ULK1 complex, which consists of Atg13, ULK1/2, FIP200, and Atg101, triggers autophagosome formation [88]. Phosphatidylinositol 3-kinase (PI3K) activation is necessary for phagophore elongation, with BECN1 inhibitors, VPS34 activators, and Beclin-1 regulators influencing this process [88]. Under enriched nutrient conditions, mTORC1 inhibits autophagy by suppressing the phosphorylation of Atg13, ULK1, and transcription factor EB (TFEB) [2]. Chronic HFD intake significantly reduces non-selective autophagy in the heart, a process associated with disruptions in the AMPK signaling pathway [59,89,90], the PI3K/Akt pathway [36], the CREG1-FBXO27-LAMP2 axis [66], the RIPK1/RIPK3 (receptor-interacting protein kinases 1/3) signaling pathway [46], and metabolism-related enzymes [34,60], all of which are discussed below (Figure 1). Both AMPK and PI3K/Akt pathways contribute to autophagy dysfunction in the hearts of STZ-induced insulin-deficient T1DM mice (Figure 2) [44,49,50]. Additionally, autophagic responses in the context of combined HFD and ischemic intervention present unique complexity [59,91].

### 3.1. AMPK Signaling Pathways Regulates Autophagy in Metabolic Cardiomyopathy

AMPK activation leads to the phosphorylation of ULK1 and translocation of the ULK complex to the ER, thereby initiating autophagy [2]. It is well-established that the SIRT1/AMPK signaling pathway is suppressed under conditions of excess nutrition [89], due to its role in regulating autophagy during caloric deprivation [90]. In obesity and T2DM, SIRT1 and AMPK are downregulated, leading to impaired autophagy [90]. In the hearts of obese mice, SIRT1 suppression contributes to autophagy inhibition via Akt signaling and regulation of autophagy-related protein (Atg) activity. Moreover, SIRT1 influences autophagy by modulating the activity of FOXO1 [9]. Impaired autophagic flux in T2DM hearts is linked to mechanisms that hinder both autophagosome formation and clearance. In T2DM hearts induced by HFD, this impairment is associated with decreased adiponectin levels. AMPK, a downstream factor activated by adiponectin, plays a role in autophagosome formation but not with clearance [59].

AMPK also contributes to pathological autophagy in the hearts of STZ-induced T1D mice [44,49]. Neuregulin-4 (Nrg4), an adipokine, has been shown to ameliorate autophagy dysfunction in T1DM cardiomyopathy by activating AMPK phosphorylation, thus enhancing autophagosome formation. This effect is blocked by AMPK inhibition [49]. These findings suggest that HFD/STZ-induced autophagy dysfunction in the heart is mediated by the suppression of AMPK signaling.

### 3.2. Inhibition of Autophagy in Metabolic Cardiomyopathy Is Accompanied by Activation of PI3K/Akt-Mediated Pathways

Glycogen synthase kinase-3 (GSK-3) and mTOR, key regulators of cardiomyocyte autophagy, are downstream targets of the PI3K/Akt pathway, which modulates cardiac hypertrophy [36]. mTOR inhibits autophagy by suppressing ULK1 activity [87]. In ob/ob mouse hearts, autophagy inhibition is accompanied by enhanced Akt/mTOR signaling, and rapamycin-mediated inhibition of mTORC1 restores cardiac autophagy [29]. In HFD-induced obese mouse hearts, increased PI3K/Akt signaling results in autophagy suppression. The loss of Akt in these hearts can rescue autophagic flux impairment through the mTOR pathway [36]. Moreover, HFD-induced autophagy defects are exacerbated by the loss of Akt2 in mouse hearts [33]. In STZ-treated mice, activation of autophagy correlates with reduced Ser9-pGSK3β and Ser473-pAKT levels, suggesting that inhibiting these proteins may promote autophagy in T1DM hearts [44].

The transcription factor nuclear factor erythroid-2 (NF-E2)-related factor 2 (Nrf2) is recognized as a key regulator of mTOR and p62/SQSTM1. Nrf2 binds to antioxidant response elements in the promoters of genes related to cellular protection and autophagy, including Atg5/7 and ULK1/2 [92]. Chronic glucolipotoxicity activates Nrf2, impairing autophagy. In STZ-treated mice with cardiomyocyte-specific knockout of the Atg5 gene, myocardial damage is reversed by Nrf2 knockout, indicating that Nrf2 contributes to autophagic dysfunction in T1DM hearts [50]. These observations suggest that inhibition of the PI3K/Akt signaling pathway plays a role in autophagy impairment in metabolic cardiomyopathy.

### 3.3. The CREG1-FBXO27-LAMP2 Axis Alleviates Diabetic Cardiomyopathy by Promoting Autophagy in Cardiomyocytes

The cellular repressor of E1A-stimulated genes 1 (CREG1), a small glycoprotein, is highly expressed in the healthy heart and has been implicated in cardiac remodeling following diabetic myocardial infarction, myocardial infarction–reperfusion (MI-R), and angiotensin-II treatment [93,94,95]. Early studies suggested that CREG1 exerts its protective effects by enhancing autophagy [66,94,95,96]. However, whether CREG1 can improve cardiac function in diabetic cardiomyopathy through autophagy regulation remains unclear.

Recent studies have demonstrated that CREG1 expression is downregulated in the hearts of diabetic mice. In T2DM models induced by 24 weeks of HFD followed by a 5-day STZ injection, autophagy dysfunction is exacerbated by CREG1 deficiency. Overexpression of CREG1 alleviates dysfunctional autophagic responses in T2DM hearts. Further investigation reveals that the CREG1-FBXO27-LAMP2 axis alleviates diabetic cardiomyopathy by promoting autophagy in cardiomyocytes [66]. Gain- and loss-of-function experiments in neonatal mouse cardiomyocytes (NMCMs) confirmed that CREG1 regulates autophagy through the upregulation of LAMP2 (lysosomal-associated membrane protein 2) expression. Moreover, they approved that FBXO27 (F-box protein 27) is involved in CREG1-mediated autophagy by interacting with LAMP2 [66]. Thus, disruption of the CREG1-FBXO27-LAMP2 axis impairs autophagy, contributing to the progression of diabetic cardiomyopathy.

### 3.4. Impaired Autophagic Flux in T2DM Is Regulated by Molecules Associated with Cell Death

Recent studies suggest that various cell death pathways, including apoptosis and necroptosis, regulate autophagy in metabolic cardiomyopathy. Receptor-interacting protein kinases 1 (RIPK1) and 3 (RIPK3) are intracellular signaling proteins that regulate necroptosis, cell survival, and inflammatory signaling [97,98]. RIPK1/RIPK3 signaling is involved in excessive autophagic flux in hearts subjected to high-glucose and -fat diets and STZ treatment. The inhibition of RIPK1/RIPK3 or the silencing of RIPK1/RIPK3 restores autophagic flux in neonatal rat fibroblasts (NRCFs) treated with high glucose and fat [46]. Silencing RIPK1/RIPK3 significantly reduces autophagy-associated proteins such as LC3-II and p62, indicating that the RIPK1/RIPK3 pathway is crucial for regulating autophagic flux in diabetic cardiomyopathy [46].

Mammalian sterile 20-like kinase 1 (Mst1), a pro-apoptotic protein, negatively regulates autophagy by phosphorylating Beclin-1 in the heart [99,100]. Mst1 phosphorylation leads to reduced interaction between Beclin-1 and BCL2, thus promoting autophagy [52,101]. In mice, ST-HFD consumption activates autophagy and reduces Mst1 and phosphorylated Mst1 levels, suggesting that the inactivation of Mst1 may trigger autophagy under early HFD conditions [41]. These findings underscore the complex interplay between autophagy and various cell death pathways in metabolic cardiomyopathy.

### 3.5. Metabolism-Related Enzymes in the Regulation of Non-Selective Autophagy in Metabolic Cardiomyopathy

Mitochondrial ALDH2 plays a vital role in maintaining autophagy homeostasis in the heart. ALDH2 expression is significantly reduced in patients with atrial fibrillation and is similarly downregulated in the hearts of obese mice. Overexpression of ALDH2 protects against HFD-induced autophagy dysfunction. Notably, the suppression of autophagy caused by chronic HFD feeding is reversed in ALDH2 transgenic mouse hearts, highlighting the enzyme’s pivotal role in preserving autophagy homeostasis [34].

Nicotinamide phosphoribosyltransferase (Nampt), the rate-limiting enzyme for nicotinamide adenine dinucleotide biosynthesis, is also crucial for cardiomyocyte metabolism. A 2021 study by Oka et al. demonstrated that HFD feeding significantly increases Nampt levels in mouse hearts, with Nampt overexpression promoting palmitate-induced autophagy in cultured cardiomyocytes [70,102]. Conversely, studies have shown that HFD impairs autophagosome turnover in the heart, which is attributed to lysosomal dysfunction. This dysfunction is mediated by the activation of Nox2, which leads to increased reactive oxygen species (ROS) production and impaired lysosomal activity via protein kinase C activation [60,103]. Notably, pharmacological inhibition of Nox2 or protein kinase C restores autophagic function in cardiomyocytes exposed to PA [60]. Collectively, these findings indicate that the abnormal regulation of metabolic enzyme caused by HFD disrupts autophagy in the heart, contributing to the pathology of metabolic cardiomyopathy.

### 3.6. Cardiac Autophagy Under HFD and Ischemic Dual Intervention

T2DM disrupts autophagic flux in cardiomyocytes following myocardial infarction [59]. While ST-HFD consumption may paradoxically improve cardiac function post-MI-R, the mechanisms underlying this protective effect remain unclear [91]. In MI-R injury, autophagic flux fails to increase in HFD-induced diabetic hearts, despite elevated LC3-II levels. This impairment is exacerbated in diabetic hearts and involves the abnormal regulation of AMPK signaling. Treatment with AdipoRon, a synthetic adiponectin analog, rescues autophagic flux in T2DM hearts post-MI-R by activating the AMPK pathway, as evidenced by increased expression of MAP1LC3, LAMP2, PIK3C3/VPS34, and BECN1. Notably, these effects are abrogated by the AMPK inhibitor compound C [59].

In non-diabetic hearts, MI-R is associated with diminished autophagosome clearance, characterized by reduced LAMP2 levels and elevated SQSTM1 accumulation. This impairment is further exacerbated in diabetic hearts [59]. However, ST-HFD exposure prior to MI-R injury may have cardioprotective effects by activating autophagy through an NF-κB-dependent pathway. Studies in wild-type mice subjected to MI-R surgery following ST-HFD (24 h to two weeks) revealed increased autophagy, which was absent in NF-κB-deficient mice, confirming the pathway’s involvement [91]. These findings suggest a dual role for HFD in regulating autophagy: while chronic HFD exacerbates autophagy dysfunction in diabetic hearts, acute HFD exposure may offer protective effects in ischemic conditions by modulating autophagy pathways.

In summary, abnormal autophagy is a critical factor in the pathogenesis of metabolic cardiomyopathy. Targeting autophagy-related pathways, including those regulated by metabolic enzymes and ischemic interventions, may provide novel therapeutic strategies for treating cardiomyopathy in chronic metabolic diseases.

## 4. Selective Autophagy in Metabolic Cardiomyopathy

Unlike non-selective autophagy, which indiscriminately degrades cytoplasmic components, selective autophagy specifically targets diverse substrates, often harmful, for degradation [13]. This process relies heavily on the recognition of cargo through ubiquitination, which not only facilitates target identification but also regulates receptor proteins involved in autophagy [13]. Various selective autophagy receptors, characterized by LC3-interacting region motifs, bind both cargo and ubiquitin, tethering them to the phagophore via interactions with LC3-family proteins [104]. Emerging evidence underscores the significant impact of HFD on selective autophagy in cardiac tissue, a topic further explored in this section (Figure 3). A comprehensive understanding of the molecular mechanisms underlying selective autophagy may inform the development of novel preventive strategies for metabolic cardiomyopathy.

### 4.1. Metabolism-Related Enzymes in the Regulation of Selective Autophagy in Metabolic Cardiomyopathy

In metabolic cardiomyopathy, mitophagy plays a pivotal role as a quality control mechanism for the removal of damaged mitochondria under various stress conditions [2]. This process is primarily mediated through two pathways: the PINK1/Parkin-dependent pathway and the receptor-mediated pathway [105,106,107]. Among these, the PINK1/Parkin pathway is one of the most extensively characterized forms of ubiquitin-mediated mitophagy [16,17]. Following mitochondrial membrane depolarization, PINK1 accumulates on the outer mitochondrial membrane, where it is stabilized and activated. Active PINK1 phosphorylates ubiquitin at Ser65, which then facilitates the recruitment and activation of Parkin to the damaged mitochondria, initiating mitophagy in a PINK1/Parkin-dependent manner [105] (Figure 3). Recent studies have identified an alternative mitophagy pathway, the Ulk1/Rab9 pathway, which can also mediate mitochondrial clearance [108]. Under stress conditions, activation of AMP-activated protein kinase (AMPK) leads to the phosphorylation of Ulk1, which in turn phosphorylates Rab9 at Ser179, triggering autophagic processes [108]. However, the precise pathological mechanisms that disrupt mitophagy homeostasis in metabolic cardiomyopathy remain poorly understood. Thus, further investigation is needed to elucidate the regulatory role of mitophagy in high-fat diet (HFD)-induced metabolic cardiomyopathy.

**Figure 3 ijms-26-01668-f003:**
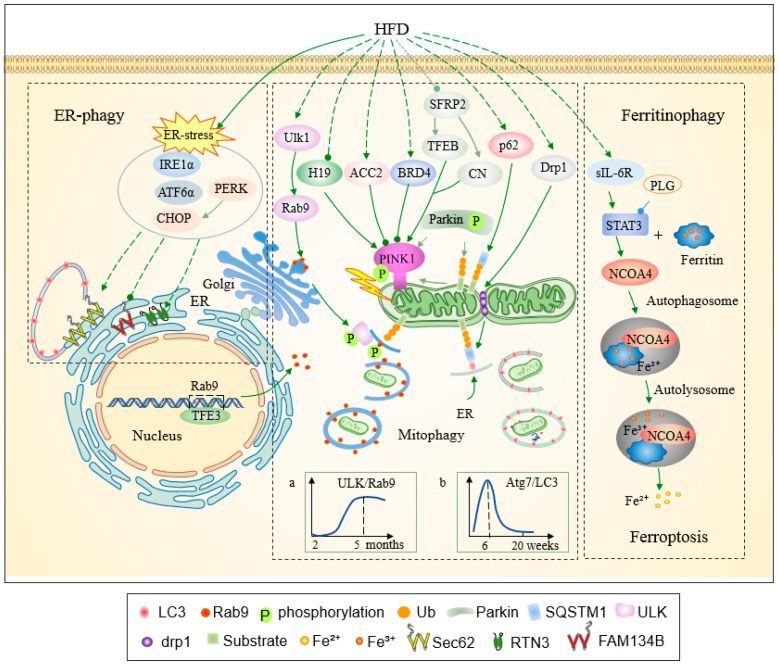
A simplified scheme for the regulation of cardiac selective autophagy in HFD-induced metabolic cardiomyopathy. ER-phagy, mitophagy, and ferritinophagy are implicated in the development HFD-induced metabolic cardiomyopathy. The PINK1/Parkin ubiquitin-mediated mitophagy pathway, which involves factors such as H19, ACC2, BRD4, SQSTM1, Nampt, and SERP2, is inhibited, and the alternative Ulk1-Rab9-dependent mitophagy pathway is activated. Panels a and b, respectively, are cited from [37,109], and depict the changes in autophagy regulated by the ULK/Rab9 and Atg7/LC3 pathways during the HFD intervention. ER stress, as evidenced by a significant increase in the protein levels of PERK, IRE1α, ATF6α and CHOP, was activated in the diabetic rat heart. Although the effect has not been demonstrated in rodent models, HG treatment has been shown to enhance ER-phagy (upregulation of Sec62 and RTN3 and downregulation of FAM134B) in H9c2 cells. This finding is corroborated by the upregulation of Sec62 and RTN3, as well as the downregulation of FAM134B. HFD significantly increases ferritinophagy by activating the STAT3/NCOA4/FTH1 axis, which promotes ferroptosis due to iron overload. Solid lines represent proven mechanisms, dotted lines represent unknown mechanisms. Targeted agents enclosed in yellow oval boxes are connected to therapeutic targets by blue lines. (↓: stimulatory modification, ⫰: inhibitory modification).

#### 4.1.1. PINK1/Parkin Pathway-Mediated Mitophagy Is Inhibited in Metabolic Cardiomyopathy

Ubiquitination-dependent mitophagy is critically regulated by the PINK1/Parkin pathway [16,17]. In the hearts of ob/ob mice, the expression of the long non-coding RNA H19 (H19) is downregulated, while PINK1/Parkin-dependent mitophagy is upregulated. H19 plays a pivotal role in metabolic regulation. Enhancing H19 expression suppresses excessive mitophagy by restricting the translation of Pink1 mRNA, thereby alleviating obesity-induced cardiomyopathy [30]. Conversely, chronic HFD consumption impairs Pink1-Parkin-mediated mitophagy [24,32]. Mitophagy appears to be initially promoted during the early phases of HFD-induced cardiomyopathy but becomes inhibited in later stages. After 20 weeks of HFD exposure, mitophagy-associated gene expression and PINK1-dependent phosphorylation are downregulated in C57 mouse hearts, suggesting the disruption of mitophagy in chronic obesity-related cardiomyopathy [32].

Recent studies reveal that HFD suppresses mitophagy through the PINK1/Parkin pathway. Simultaneously, HFD significantly increases the expression of cardiac bromodomain-containing protein 4 (BRD4), which inhibits PINK1/Parkin-mediated mitophagy. Treatment with JQ1, a BRD4 inhibitor, restores PINK1/Parkin-mediated mitophagy and attenuates HFD-induced diabetic cardiomyopathy, highlighting the regulatory role of BRD4 in this pathway [110]. Shao et al. further demonstrated that long-term high-fat diet (LT-HFD) treatment decreases mitophagy in cardiomyocytes [24]. This suppression of PINK1/Parkin-mediated mitophagy was linked to the upregulation of acetyl-CoA carboxylase 2 (ACC2). In isolated cardiomyocytes, mitophagy activity was significantly reduced following prolonged HFD treatment. However, in ACC2-deficient mouse hearts, mitophagy remained unaffected, indicating that the absence of ACC2 mitigates the reduction in Parkin levels within mitochondrial fractions observed during chronic HFD exposure. Interestingly, while fasting or treatment with carbonyl cyanide p-trichloromethoxyphenyl hydrazone (CCCP) typically enhances Parkin-mediated mitophagy, this effect is inhibited by chronic HFD feeding, but not in ACC2-deficient mice [24]. These findings underscore ACC2’s role in dampening mitophagy activity during HFD-induced metabolic stress.

Collectively, these results indicate that PINK1/Parkin-mediated mitophagy is inhibited in chronic HFD-induced metabolic cardiomyopathy but is paradoxically upregulated in ob/ob mouse hearts. This discrepancy may reflect fundamental differences between genetically induced obesity and diet-induced obesity. The inhibition of PINK1/Parkin-mediated mitophagy in HFD-induced cardiomyopathy is primarily regulated by BRD4 and ACC2, suggesting potential therapeutic targets to modulate mitophagy and mitigate disease progression.

#### 4.1.2. Ulk1-Rab9-Dependent Alternative Mitophagy Activation During Chronic HFD Consumption

Autophagy can be classified into two categories based on the source of the autophagosome membrane: conventional autophagy, which is Atg7/LC3-dependent, and alternative autophagy, which relies on Ulk1/Rab9. Recent studies have demonstrated that Atg7/LC3-dependent mitophagy is predominantly activated during the early stages of HFD consumption, while prolonged HFD exposure suppresses this pathway and activates Ulk1/Rab9-mediated mitophagy. Specifically, Tong et al. in 2023 showed that after 12 weeks of HFD intervention, the Ulk1/Rab9 pathway became the primary mechanism for mitochondrial clearance in the heart, compensating for the decline in Atg7/LC3-dependent mitophagy [55].

In cardiac-specific Ulk1 conditional knockout (cKO) mice, the loss of Ulk1-mediated mitophagy was accompanied by increased reliance on Atg7- and Parkin-mediated mitophagy to mitigate mitochondrial damage during HFD consumption [37]. Atg7 and Parkin were essential for maintaining cardiac function under HFD stress by removing damaged mitochondria. In Atg7 cKO and Parkin knockout mice, mitophagy was severely impaired, leading to mitochondrial damage accumulation and exacerbated cardiac dysfunction [41]. The GTPase dynamin-related protein 1 (DRP1) also plays a critical role in mitochondrial fission and autophagy regulation [111]. In cardiac-specific DRP1 knockout mice, mitophagy and mitochondrial quality control were significantly compromised, evidenced by a reduced mitochondrial DNA/nuclear DNA ratio and impaired Atg7-dependent autophagy [55].

Notably, DRP1 facilitates the upregulation of Atg7/LC3-dependent mitophagy during the early phases of HFD consumption, as revealed by GFP-LC3 labeling experiments [55]. However, during chronic HFD exposure, Ulk1/Rab9-dependent mitophagy becomes the dominant pathway. Rab9 phosphorylation at serine 179 by Ulk1 is essential for this alternative mitophagy process, which protects mitochondrial integrity and cardiac function [112]. DRP1 is also implicated in regulating Ulk1- and Rab9-mediated mitophagy, further underscoring its role in the heart’s adaptive response to chronic HFD-induced stress [55].

Chronic HFD consumption disrupts mitochondrial homeostasis and induces structural disorganization of mitochondrial cristae, contributing to cardiac hypertrophy. Cardiac-specific deletion of Ulk1 or inhibition of Rab9 phosphorylation exacerbates these adverse effects by suppressing alternative mitophagy [37,108]. Tong et al. demonstrated increased co-localization of mitochondria and Rab9 in cardiomyocytes from cardiac-specific YFP-Rab9 transgenic mice following 12 weeks of HFD, providing evidence of enhanced alternative mitophagy [37].

Additionally, the transcription factor TFE3 (transcription factor binding to IGHM enhancer 3) has been identified as a critical regulator of Rab9-dependent mitophagy during chronic HFD consumption [55]. HFD significantly increases TFE3 expression, and chromatin immunoprecipitation assays confirmed enhanced TFE3 binding to the Rab9 promoter after 20 weeks of HFD. The deletion of TFE3 in the heart impaired Rab9-dependent mitophagy, while Rab9 overexpression in transgenic mice (Tg-Rab9) mitigated mitochondrial dysfunction caused by HFD through enhanced alternative mitophagy [37].

Importantly, Rab9-dependent alternative mitophagy can compensate for insufficient mitophagy resulting from Atg7 deficiency, as demonstrated in Atg7 cKO mice with Rab9 overexpression [37]. These findings highlight the critical role of Ulk1/Rab9-dependent mitophagy in maintaining mitochondrial quality and cardiac function during chronic HFD consumption, with TFE3 serving as a key regulatory factor.

#### 4.1.3. SFRP2 and TB1 in the Regulation of Mitophagy in HFD-Induced Metabolic Cardiomyopathy

Secreted frizzled-related protein 2 (SFRP2), a novel adipokine, has demonstrated cardioprotective effects in diabetic cardiomyopathy [113]. Recent studies have revealed that SFRP2 can activate mitophagy in the diabetic heart, thereby contributing to its cardioprotective effects [48]. In H9c2 cells, a glucolipotoxic environment significantly downregulated the protein levels of SFRP2. Under these conditions, overexpression of SFRP2 in H9c2 cells resulted in the upregulation of transcription factor EB (TFEB) and calcineurin expression, leading to the activation of mitophagy. These findings suggest that SFRP2 positively regulates mitophagy through the calcineurin/TFEB pathway [48].

Similar results were observed in diabetic rats generated by intraperitoneal administration of STZ following 4 weeks of HFD feeding. The myocardial expression of SFRP2 was significantly decreased in HFD-induced diabetic rats. Furthermore, mitophagy in the hearts of these rats was impaired, but this dysfunction was significantly alleviated by overexpression of SFRP2 via AAV-Sfrp2. SFRP2 overexpression promoted mitophagy in diabetic rats by targeting the calcineurin/TFEB pathway [48].

Tat-Beclin1 (TB1), an effective autophagy-inducing peptide derived from Beclin1, has been shown to selectively bind to GAPR1, thereby inhibiting GAPR1’s negative regulation of Beclin1 and promoting autophagy [114]. Further studies have demonstrated that TB1 treatment improves diabetic cardiomyopathy by enhancing mitophagy. After 2 weeks and 3 months of HFD feeding, TB1 administration increased LC3II levels in mitochondrial fractions and rescued mitophagy and cardiac dysfunction associated with HFD consumption. Notably, TB1 also attenuated cardiac dysfunction and mitochondrial abnormalities in Parkin knockout mice. Additionally, nuclear respiratory factor 1, a transcription factor involved in mitochondrial biogenesis, was significantly upregulated following TB1 treatment. In mitotimer transgenic mice, which track mitochondrial turnover, a decrease in mitochondrial turnover was observed after HFD consumption. TB1 effectively rescued the reduced mitochondrial turnover rates [41].

In conclusion, these findings indicate that mitophagy is impaired in HFD-induced metabolic cardiomyopathy, primarily due to disruptions in the PINK1/Parkin pathway and the activation of the Ulk1/Rab9 pathway. While PINK1/Parkin-mediated mitophagy is upregulated in the hearts of ob/ob mice [37,55], conventional autophagy in the heart, which is dependent on Atg7/LC3, is typically initiated in the early stages of HFD consumption. However, this process becomes inhibited following LT-HFD intervention. In contrast, Ulk1/Rab9-dependent alternative mitophagy is activated in the hearts of mice after prolonged HFD exposure [55]. Moreover, SFRP2 and TB1 play key roles in the regulation of mitophagy in HFD/STZ-induced metabolic cardiomyopathy. The role of mitophagy in the development of metabolic cardiomyopathy is complex and remains a critical area of research, warranting further studies to elucidate its mechanisms in detail.

### 4.2. Ferritinophagy Is Activated by the STAT3/NCOA4/FTH1 Axis in HFD-Induced Metabolic Cardiomyopathy

Under physiological conditions, ferritinophagy is a crucial mechanism for maintaining iron homeostasis [115]. NCOA4 binds to ferritin heavy chain ferritin1 (FTH1) and transports it to autolysosomes for degradation [115]. This process plays a pivotal role in ferroptosis by regulating intracellular iron balance and modulating ROS production. In the context of metabolic cardiomyopathy, the ferroptosis-related pathological process may be mitigated by targeting ferritinophagy. A positive correlation has been observed between ferritinophagy and ferroptosis, suggesting that ferritinophagy can promote ferroptosis through iron overload [116]. Recent advances have underscored the potential contribution of ferroptosis to metabolic cardiomyopathy [80,81,117,118], with emerging evidence highlighting the central role of ferritinophagy in HFD/STZ-induced metabolic cardiomyopathy [50,57].

HFD treatment has been shown to activate the STAT3/NCOA4/FTH1 axis in the mouse heart. Along with the increased levels of ferric iron in the HFD-induced mouse heart, HFD treatment significantly enhanced the expression of NCOA4, FTH1, PTGS2, and p-STAT3 in heart tissue [57]. These findings suggest that HFD significantly promotes ferritinophagy through the activation of the STAT3/NCOA4/FTH1 axis. Consistent with the in vivo findings, oleic acid/palmitic acid (OA/PA) treatment also activated the STAT3/NCOA4/FTH1 axis in H9C2 cells [57]. HFD-induced obesity cardiomyopathy is associated with ferroptosis activation [80,81,82]. In line with these in vivo findings, OA/PA treatment induced cellular injury in H9C2 cells and activated ferroptosis. Ferroptosis inhibitors, such as ferrostatin-1 and 3-methyladenine, alleviated cellular injury induced by OA/PA. The knockdown of NCOA4 normalized the abnormal expression of FTH1 and PTGS2 and mitigated excessive ROS production in OA/PA-treated cardiomyocytes [57]. Furthermore, Stattic, a STAT3 inhibitor, or STAT3 siRNA prevented the upregulation of proteins involved in ferritinophagy (NCOA4, FTH1, LC3, Atg5, and Atg7), indicating that STAT3 plays an indispensable role in OA/PA-induced ferritinophagy [57].

T1DM triggers glucolipotoxicity, which impairs autolysosome function and intensifies Nrf2-driven lipid peroxidation [93], thus contributing to ferroptosis in cardiomyocytes and exacerbating diabetic cardiomyopathy [50]. In STZ-induced T1DM mice, cardiac iron deposition increased 9 months after the onset of diabetes, with a further enhancement in iron deposition following the loss of Atg5. Ferritinophagy appears to be activated in response to T1DM in STZ-induced diabetic cardiomyopathy. Erastin, a ferroptosis inducer, plays a role in Nrf2-mediated ferroptosis/ferritinophagy in the T1DM mouse heart by increasing the expression of FTH1 and ferritin light chain 1, thereby inducing iron storage in cardiomyocytes [50].

In conclusion, these findings suggest novel prospective targets for the treatment of metabolic cardiomyopathy in the near future [50,57]. The precise mechanism by which ferritinophagy contributes to the development of metabolic cardiomyopathy remains unclear. Therefore, further research is required to accelerate the development of targeted therapies that modulate NCOA4-mediated ferritinophagy to address ferroptosis in metabolic cardiomyopathy.

### 4.3. ER-Phagy Is Activated in Diabetic Cardiomyopathy

As a key selective autophagic process, ER-phagy serves to degrade dysfunctional ER membranes that are caused by ER stress. To date, several autophagy receptors involved in ER-phagy have been identified in mammals, including FAM134B (reticulophagy regulator 1), SEC62 (SEC62 homolog), RTN3 (reticulon 3), CCPG1 (cell-cycle progression gene 1), ATL3 (atlastin 3), TEX264 (testis expressed 264), and p62 [13]. Among these, Sec62, FAM134B, RTN3, and CCPG1 play regulatory roles as ER-phagy receptors [119,120,121]. The ER undergoes continuous remodeling through the process of ER-phagy, ensuring cellular homeostasis by eliminating damaged or excessive ER components [119].

Fumagalli et al. demonstrated that Sec62, an ER-resident autophagy receptor, can independently induce ER-phagy through its interaction with LC3 [120]. Sec62 plays a crucial role in recovering from ER stress by selectively delivering ER components to the autophagic system for degradation [119]. More recently, a study by González et al. revealed that the ubiquitination of FAM134B within its reticulon homology domain is pivotal for activating ER-phagy [119]. This process promotes the clustering of FAM134B receptors and their interaction with lipidated LC3B, highlighting FAM134B’s critical role in ER-phagy [119]. The discovery of multiple ER-phagy receptors has spurred further studies into the roles of ER-phagy in cellular stress responses.

The process of ER-phagy has been implicated in the development of diabetic cardiomyopathy. ER stress, which regulates the expression and phosphorylation levels of ER-phagy receptors, is an important contributor to the progression of diabetic cardiomyopathy [122]. Evidence suggests that obesity induces ER stress in the hearts of mice, as shown by the upregulation of eIF2α (eukaryotic initiation factor 2, α subunit), GRP78 (Glucose-regulated protein, 78 kDa), and ATF4 (activating transcription factor 4). Additionally, palmitic acid has been found to induce ER stress in cultured cardiomyocytes [123]. Rani et al. demonstrated that ER-phagy is activated in diabetic cardiomyopathy through both in vitro and in vivo experiments. ER stress, indicated by a significant increase in the protein levels of PERK (protein kinase RNA-like endoplasmic reticulum kinase), IRE1α (inositol requiring enzyme 1α), ATF6α (activating transcription factor 6α), and CHOP (C/EBP homologous protein), was activated in the hearts of diabetic rats. Moreover, ER-phagy, evidenced by the upregulation of Sec62 and RTN3 and the downregulation of FAM134B, was enhanced in H9c2 cells under high glucose treatment. Furthermore, HG exposure significantly upregulated the protein levels of ER stress markers and promoted ER stress-mediated apoptosis in H9c2 cardiomyocytes [58].

Contemporary research into selective autophagy mechanisms in metabolic cardiomyopathy includes mitophagy [24,32], ferritinophagy [57], and ER-phagy [58]. Conventional autophagy, dependent on Atg7/LC3, is activated in the early stages of HFD consumption [37], while Ulk1/Rab9-dependent alternative mitophagy is activated in the mouse heart during LT-HFD intervention [55]. Although the precise ways in which ferritinophagy and ER-phagy exacerbate metabolic cardiomyopathy remain unknown, these emerging studies highlight the involvement of these processes.

Although the mechanisms by which ER-phagy contributes to diabetic cardiomyopathy remain incomplete, these findings provide novel insights into the potential therapeutic targeting of ER-phagy. Future research is needed to elucidate the precise mechanisms of ER-phagy and its role in the pathophysiology of diabetic cardiomyopathy. A comprehensive understanding of how ferritinophagy and ER-phagy function could aid in designing novel strategies for the prevention and treatment of metabolic cardiomyopathy.

## 5. Autophagy as a Possible Therapeutic Target for Metabolic Cardiomyopathy

Current data support the notion that abnormal regulation of both non-selective and selective autophagy plays a pivotal role in the pathogenesis of metabolic cardiomyopathy [9,10]. The restoration of autophagic function via pharmacological interventions has shown promising results in alleviating cardiac injury in experimental models of metabolic cardiomyopathy [9]. A range of plant extracts, exogenous organic compounds, and endogenous substances have been explored for their potential to target autophagy [10], thereby alleviating the detrimental effects of metabolic cardiomyopathy.

### 5.1. Plant Extracts as Therapeutic Agents Targeting Autophagy in Metabolic Cardiomyopathy

In diabetic cardiomyopathy, compounds such as cinacalcet (Cin) and curcumin stimulate autophagy via the AMPK pathway. In HFD- and STZ-induced T2DM rat models, autophagic dysfunction was observed, with decreased autophagy-related proteins in the heart. Cinacalcet restores cardiac function by activating autophagy through AMPK signaling, protecting cardiomyocytes from apoptosis and reducing myocardial injury [101]. Curcumin also promotes autophagy and reduces apoptosis in T1DM models by activating AMPK and JNK1 pathways [52]. Additionally, dihydromyricetin (DHM), by downregulating miR-34a, enhances autophagy and shows therapeutic potential in diabetic cardiomyopathy [53]. Piperlongumine, derived from Piper species, targets ferritinophagy, improving cardiac hypertrophy, injury, and fibrosis in HFD-induced mice by inhibiting STAT3 phosphorylation and ferritin overload [57]. Polydatin, from Polygonum cuspidatum, promotes autophagy and activates sirtuin 3 in STZ-induced diabetic mice [47]. Berberine, derived from Coptis chinensis, alleviates impaired autophagic flux by binding to Drp1 in heart failure models induced by HFD and L-NAME [76]. These compounds represent promising therapeutic agents targeting both non-selective and selective autophagy in metabolic cardiomyopathy, warranting further investigation to optimize their clinical application.

### 5.2. Beneficial Effects of Endogenous Substances on Autophagy in Metabolic Cardiomyopathy

In 2022, Comella et al. reported that oleoylethanolamide (OEA), an endogenous peroxisome proliferator-activated receptor alpha agonist, mitigates HFD-induced obesity cardiomyopathy by restoring autophagic function and normalizing the levels of inflammatory cytokines, chemokines, and pro-inflammatory mediators [31]. Additionally, the expression of meteorin-like (Metrnl) was found to be significantly reduced in both STZ-induced T1DM mouse hearts and db/db T2DM mouse hearts. Metrnl has been shown to exert a cardioprotective effect by activating autophagy through the LKB1/AMPK/ULK1 signaling pathway. In primary cardiomyocytes exposed to high glucose, autophagy was suppressed; however, this suppression was reversed by Metrnl. Prolonged high glucose exposure led to a significant reduction in the phosphorylation of AMPK, LKB1, and ULK1 in these cells. Notably, the beneficial effects of Metrnl on autophagy were abolished upon inhibition of AMPK, further confirming the importance of the LKB1/AMPK pathway in the Metrnl-mediated regulation of autophagy [54]. LKB1, a key upstream regulator of AMPK, was found to play a crucial role, as Metrnl-induced phosphorylation and activation of AMPK were significantly diminished in the absence of LKB1 [54]. This phenomenon suggests that Metrnl regulates autophagy in an LKB1/AMPK-dependent manner.

As previously discussed, different studies have shown that autophagy can have both protective and pathogenic effects in the development of STZ-induced T1DM cardiomyopathy. The dual nature of autophagy in this context indicates that its effects may vary depending on the specific stage of the disease and the underlying conditions. Zhang et al. demonstrated that excessive autophagy in STZ-induced insulin-deficient mice contributes to myocardial injury, a process that was ameliorated by treatment with angiotensin IV (Ang IV) [45]. In these mice, the number of autophagosomes, as indicated by monodansylcadaverine staining, was significantly higher in the myocardium compared to normal controls. Gene ontology enrichment analysis revealed that the mTOR and FoxO pathways were most enriched in the context of autophagy in diabetes mellitus. Ang IV mitigated excessive autophagy in STZ-induced T1DM cardiomyopathy, partly by inhibiting the FoxO1 signaling pathway. Moreover, the cardioprotective effects of Ang IV were replicated by the FoxO1 inhibitor AS, while the AT4 receptor antagonist divalinal prevented the beneficial effects of Ang IV [45].

In ob/ob mouse hearts, the long non-coding RNA H19 has been shown to inhibit excessive PINK1/Parkin-dependent mitophagy by restricting the translation of Pink1 mRNA, thereby ameliorating obesity-related cardiomyopathy [30]. Additionally, in cardiomyocytes challenged with palmitic acid for 24 h, mitophagy, as indicated by Mito-Keima, was decreased. However, this decrease in mitophagy was restored by the administration of Urolithin A. Urolithin A, a naturally occurring metabolite derived from gut microbiota, enhances mitophagy in HFD-induced metabolic cardiomyopathy by upregulating the expression of Parkin and PINK1, key mediators of mitophagy [32].

### 5.3. Lipid-/Glucose-Lowering Drugs Ameliorate Diabetic Cardiomyopathy by Regulating Autophagy

Fenofibrate, an exogenous PPARα agonist, has been shown to enhance cardiac autophagy via the FGF21/SIRT1 pathway, thereby alleviating STZ-induced metabolic cardiomyopathy [51]. Numerous studies have demonstrated that sodium–glucose cotransporter-2 (SGLT2) inhibitors improve diabetic cardiomyopathy by correcting abnormal autophagy. Empagliflozin (EMPA), a highly selective SGLT2 inhibitor, is a widely used anti-diabetic drug [124]. T2DM is often characterized by suppression of SIRT1 and AMPK, as well as impaired autophagy. SGLT2 inhibitors, such as EMPA, protect the heart in T2DM by restoring autophagy through the activation of SIRT1/AMPK and suppression of the Akt/mTOR signaling pathway [90].

Aragón-Herrera et al. reported that EMPA rescued impaired autophagy in atrial tissues of ZDF rats, enhancing cardiac autophagy by activating AMPK signaling and reducing cardiac CD36 expression in these animals [28]. Madonna et al. also demonstrated that EMPA improves cardiac function in STZ-induced insulin-deficient mice by modulating autophagy. In these mice, the protein levels of SGLT2, LC3-II, and SQSTM1 were upregulated, and EMPA reversed these effects, potentially through the GSK3β/Akt pathway. Specifically, EMPA improved the downregulation of Ser9-pGSK3β and Ser473-pAKT induced by STZ in the hearts of type 1 diabetic mice [44]. Furthermore, EMPA has been shown to regulate excessive autophagy by directly inhibiting Na^+^/H^+^ exchanger 1 activity in cardiomyocytes [125].

The involvement of abnormal autophagy in the pathogenesis and progression of metabolic cardiomyopathy, including obesity-related cardiomyopathy and diabetic cardiomyopathy, is now well established. The detailed molecular mechanisms underlying both non-selective and selective autophagy in metabolic cardiomyopathy have been described above. This section highlights how plant extracts [47,101], endogenous substances [30,31,32,45,54], and lipid-/glucose-lowering drugs [28,51] can improve autophagy in metabolic cardiomyopathy. Therefore, targeting autophagy may represent a promising strategy for attenuating metabolic cardiomyopathy. Together, these findings suggest that the lipid-lowering drug, such as fenofibrate, and the glucose-lowering drug, such as EMPA, can improve cardiac function by correcting abnormal autophagy, offering potential therapeutic avenues for treating metabolic cardiomyopathy.

### 5.4. Caloric Restriction and Physical Exertion Prevent Metabolic Cardiomyopathy in Human

Over the past century, the focus of human health challenges has shifted from diseases caused by insufficient nutrition to those associated with overnutrition, such as obesity, diabetes, and cardiometabolic disorders. Research indicates that intermittent fasting and long-term caloric restriction can activate a heart-healthy metabolic program and enhance autophagic activity, which may protect against metabolic cardiomyopathy [126]. Autophagy plays a pivotal role during caloric restriction and physical exertion, aiding cellular survival under nutrient-deficient conditions, while its activity is suppressed under nutrient-rich conditions [127].

Fasting strategies, particularly intermittent fasting and long-term caloric restriction, have been shown to activate autophagy and improve metabolic health. Fasting in humans has been reported to mitigate obesogenic cardiomyopathy by enhancing non-selective autophagy [128]. Research by Hofer et al. in human volunteers demonstrated that eukaryotic translation initiation factor 5A hypusination (eIF5AH) was upregulated in peripheral blood mononuclear cells (PBMCs) following fasting, whereas total eIF5A levels remained unchanged [129]. Subsequent studies revealed that eIF5AH is involved in fasting-induced autophagy enhancement [130]. Furthermore, long-term therapeutic fasting (with a daily caloric intake of around 250 kcal) significantly increases serum spermidine (SPD) levels, which enhances autophagic processes [129]. eIF5A is crucial in mediating Atg8-family protein lipidation and autophagosome formation [131]. Mechanistically, SPD drives fasting-induced autophagy by hypusinating eIF5A, thereby facilitating the autophagic process [129].

Exercise, particularly endurance and resistance training, also plays a key role in regulating autophagy in human skeletal muscle and the cardiovascular system. Endurance exercise activates autophagy markers in human skeletal muscle within the first 2 h of recovery, and 8 weeks of training enhances the regulation of autophagy and mitophagy [132]. In addition to endurance exercise, Estebanez et al. showed that 8 weeks of resistance exercise training could inhibit mitophagy activation in PBMCs from healthy elderly individuals [133]. Moreover, studies by Park et al. demonstrated that rhythmic handgrip exercise significantly upregulated autophagy proteins in arterial endothelial cells, accompanied by the increased phosphorylation of endothelial nitric oxide synthase (eNOS) and enhanced nitric oxide (NO) generation. These findings suggest that the eNOS/NO signaling pathway plays an important role in exercise-induced autophagy regulation [134].

When compared to pharmacological interventions, such as plant extracts, endogenous substances, and lipid-/glucose-lowering drugs, caloric restriction and physical exertion target autophagy through more natural, physiological pathways (Figure 4). These lifestyle interventions offer potential advantages in terms of sustainability and reducing adverse side effects. They also represent a more holistic approach, addressing both metabolic and cardiovascular health through lifestyle modifications. However, their effectiveness may vary depending on individual health conditions and the extent of metabolic dysfunction. Therefore, integrating caloric restriction and physical exertion with other autophagy-targeting therapies may provide a comprehensive strategy to prevent and manage metabolic cardiomyopathy.

## 6. Conclusions and Perspective

In conclusion, research emphasizes the critical roles of both non-selective (Figure 1 and Figure 2) and selective autophagy (Figure 3) in HFD/STZ-induced metabolic cardiomyopathy. HFD-induced nutrient overload suppresses non-selective autophagy, while selective processes like ferritinophagy and ER-phagy are upregulated under metabolic stress [57,58]. Key pathways involved in non-selective autophagy include energy-sensing mechanisms [36,90], the CREG1-FBXO27-LAMP2 axis [66], and enzymes-regulating metabolism [34,70]. Mitophagy, particularly the Ulk1/Rab9-dependent pathway, plays a crucial role, with LT-HFD inhibiting the Atg7/LC3-dependent mitophagy and activating the alternative pathway [37,55]. However, the molecular mechanisms of selective autophagy processes remain largely unclear.

Recent studies highlight the potential of autophagy-targeting therapies, including plant extracts [47,101], endogenous substances [30,31,32,45,54], and lipid- and glucose-lowering drugs [28,51], to mitigate metabolic cardiomyopathy. While autophagy is essential for cardiac homeostasis, alterations in its flux are common in metabolic cardiomyopathy. Several signaling pathways of non-selective autophagy are understood, but many regulatory factors and pathway interactions remain unknown. Recent progress in lipophagy research, particularly its role in lipid metabolism and its potential impact in diseases like T2DM and liver models, expands our understanding [135,136]. However, the role of cardiac lipophagy in metabolic cardiomyopathy is still unexplored.

The activation of PPARβ/δ has been shown to alleviate the detrimental effects of endoplasmic reticulum (ER) stress induced by saturated fatty acids by promoting autophagy in AC16 cells, a human cardiomyocyte cell line [38]. Currently, autophagy levels in the human heart are typically modeled using human-induced pluripotent stem cell-derived cardiomyocytes (iPSC-CMs) from patients [137,138,139,140]. The PLEKHM2 gene, which encodes an autophagy regulator, is mutated in individuals with severe recessive dilated cardiomyopathy. In iPSC-CMs derived from patients with PLEKHM2 mutations, autophagic activity is notably reduced [137]. Similarly, mutations in the gene-encoding LAMP-2 lead to impaired autophagy in Danon cardiomyopathy, and autophagy assessments are conducted using iPSC-CMs [138]. Despite these advancements, the mechanisms underlying autophagy in the human heart in the context of metabolic cardiomyopathy remain largely unexplored. Detecting cardiac autophagy status is essential for selecting appropriate therapeutic interventions; however, assessing autophagy levels and autophagic flux in vivo in the human heart remains challenging [10,141].

## Figures and Tables

**Figure 1 ijms-26-01668-f001:**
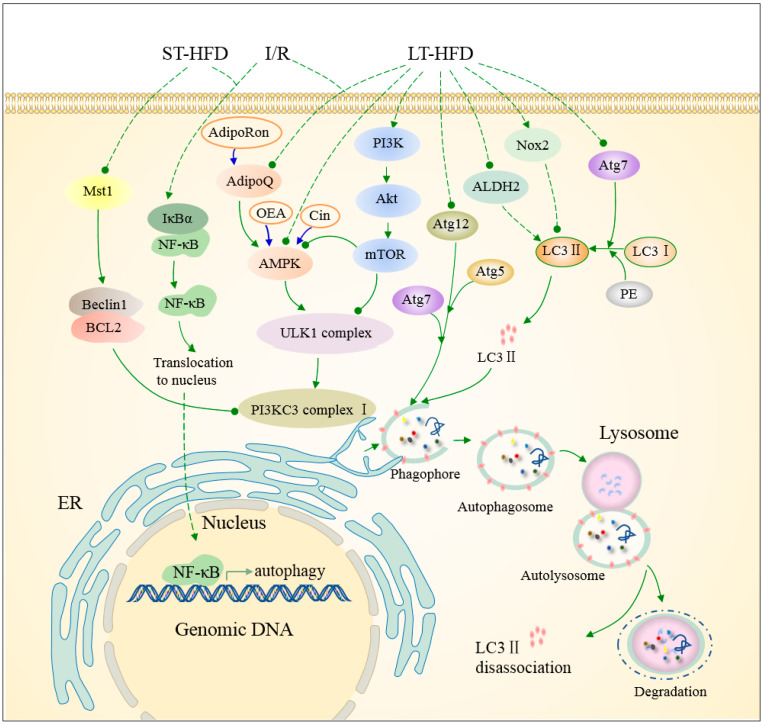
A comprehensive scheme for the complex regulation of cardiac non-selective autophagy in HFD-induced metabolic cardiomyopathy. LT-HFD (long-term high-fat diet) consumption not only activates the mTOR signaling pathway but also suppresses the AMPK signaling pathway in the heart, which impairs the initiation of phagophore nucleation. Vesicle expansion and autophagosome formation guided by ATG5-ATG12 and ATG7-LC3 was inhibited in HFD-induced metabolic cardiomyopathy. Metabolism-related enzymes included aldehyde dehydrogenase (ALDH2) and NADPH oxidase 2 (Nox2) are involved in the regulation of autophagy in metabolic cardiomyopathy. Short-term high-fat diet (ST-HFD) consumption prior to I/R promoted autophagy by the NF-kB-dependent signaling pathway. Otherwise, autophagy was activated by ST-HFD consumption in mouse hearts by inactivation of Mst1. Solid lines represent proven mechanisms, dotted lines represent unknown mechanisms. Targeted agents enclosed in yellow oval boxes are connected to therapeutic targets by blue lines. Red dots indicate LC3 II. The rest of diverse color dots indicates non-selective components involved. (↓: stimulatory modification, ⫰: inhibitory modification). (Not to scale).

**Figure 2 ijms-26-01668-f002:**
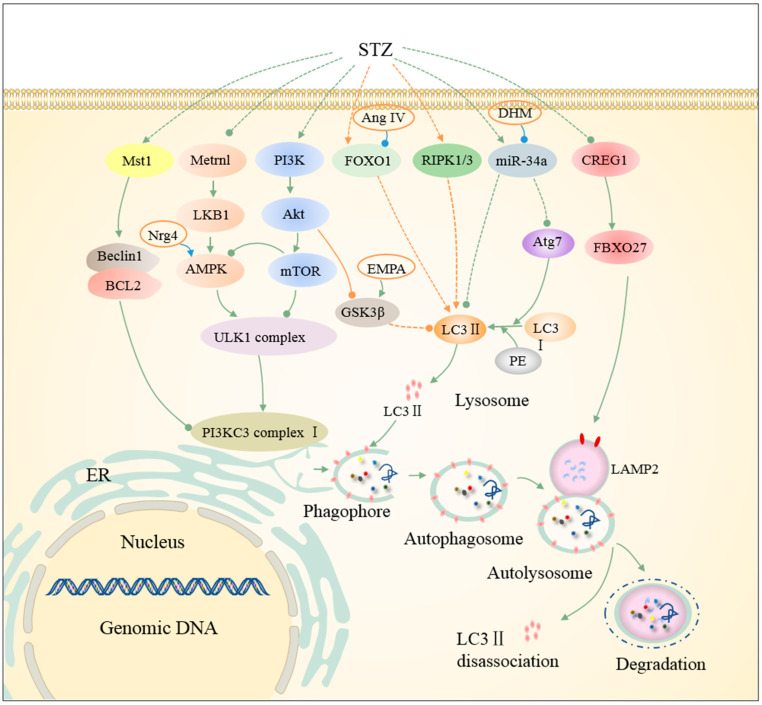
A comprehensive scheme for the complex regulation of cardiac non-selective autophagy in STZ-induced metabolic cardiomyopathy. Various studies have indicated that autophagy plays both protective and pathogenic roles in the development of STZ-induced metabolic cardiomyopathy. The pathways involved in the downregulation of autophagy STZ-induced metabolic cardiomyopathy are the LKB1/AMPK/ULK1 signaling pathway, the PI3K/AKT/mTOR signaling pathway, the axis of CREG1-FBXO27-LAMP2, and the AKT/GSK3β signaling pathway. Mst1 suppresses autophagy by increasing interaction between Beclin1 and BCL2. FOXO1 (forkhead box protein O1) induced excessive autophagy in STZ-induced metabolic cardiomyopathy. The RIPK1/RIPK3 signaling pathway and FOXO1 was involved in excessive autophagic flux in the heart, which was induced by STZ. Hyperactivation of miR-34a promotes autophagy in STZ-induced metabolic cardiomyopathy. Solid lines represent proven mechanisms, dotted-line arrows represent unknown mechanisms. Targeted agents enclosed in yellow oval boxes are connected to therapeutic targets by blue lines. Red dots indicate LC3 II. The rest of diverse color dots indicates non-selective components involved. (↓: stimulatory modification, ⫰: inhibitory modification). (Not to scale).

**Figure 4 ijms-26-01668-f004:**
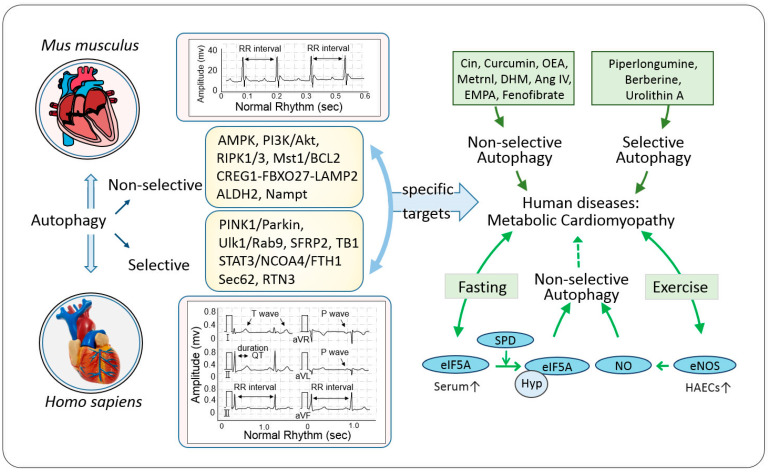
Mechanistic insights and multifaceted applications of autophagy as a potential therapeutic target for metabolic cardiomyopathies. This illustration depicts the intricate autophagy relationship, and operates through diverse pathways separately associated with non-selective and selective autophagy. Autophagy mechanisms ameliorate the positive effects of fasting and exercise on metabolic cardiomyopathy in humans. Solid lines represent confirmed mechanisms, while dotted lines represent possible mechanisms. Blue arrows are physiological paths; Dark green single-head arrows, selective and non-selective autophagy, metabolic cardiomyopathy; Light green double-head arrows, interference improvement of metabolic cardiomyopathy; Light green single-head arrows, non-selective autophagy fed back to improvement of metabolic cardiomyopathy.

**Table 1 ijms-26-01668-t001:** Autophagy and its associated genes in HFD/STZ-induced metabolic cardiomyopathy.

Types	Models	Organisms	Autophagy Genes in the Cardiac Function	References
Non- selective autophagy	HFD	C57, FVB mice	LC3-II/I ↓	[31,33,35,36,37,38,39,59,60]
		C57 mice	Atg5, Atg12 and Beclin-1 ↓	[35,38,39,61,62]
		FVB mice	Atg7 ↓	[34,36]
		FVB mice, C57 mice	p-AMPK ↓, and p-mTOR ↑	[34,36,63,64]
	STZ	SD rats, Wistar rats, C57 mice	LC3-II/I ↓	[47,48,52,53,65]
		SD rats, C57 mice	LC3-II/I and active cathepsin D ↑	[23,44,46,49,50,51,66]
		C57 mice, Wistar rats	Atg7 and Beclin-1 ↓	[47,49,53,65,66,67]
		C57 mice	p-mTOR and ULK1 ↑	[49,52,68,69]
	in vitro	H9C2 cells, NRCMs, NMCMs	LC3-II/I ↓	[47,50,51,52,53,54,65]
		NRCMs, NMCMs	Atg7 and Beclin-1 ↓	[47,53,65,67]
		H9C2 cell, NRCFs, NRCMs	LC3 ↑, Atg5, and Atg7 ↑	[45,46,57,70]
Selective autophagy	HFD	Mito-Keima mice (FVB), C57 mice	Mito-LC3 II, mito-P62, and Parkin ↓	[24,32,71,72]
		Mito-Keima mice (C57)	Mito-LC3 II and mito-p-Ulk1 ↑	[41] *
		C57 mice	Pink1, Rab7, Mfn1, Atg9, and mito-p-S65-Ub ↓; Rab9, Drp1, TOM20, and TIM23 ↑	[32,33,41,72,73,74,75,76,77,78,79,80]
		C57, FVB, KK-Ay/J mice	SQSTM1/p62 ↑	[33,34,39,40]
		C57 mice, ApoE^−/−^ mice	PTGS2 ↑	[57,81]
		SD rats, ApoE^−/−^ mice	FTH1 ↓	[57,82]
		ApoE^−/−^ mice	NCOA4 and p-STAT3 ↑	[57,83]
		Wistar rats	GRP78, PERK, IRE1α, ATF6α, and CHOP ↑	[58,84]
	in vitro	H9C2, NRCMs, NMCMs, NRCFs	SQSTM1/p62 ↑	[46,47,51,52,53,54,65]
		H9C2, NRCMs	PINK1, Parkin, mito-LC3, and SQSTM1/p62 ↓	[24,32,45]
		H9C2	Sec62, RTN3, and NCOA4 ↑; FTH1 and FAM134B ↓	[57,58,85,86]

Note: The up arrow (↑) indicates upregulation, which refers to the increasing expression of proteins, while the down arrow (↓) indicates downregulation. An asterisk (*) denotes significant differences in the levels of LC3 II in mitochondria, attributable to varying durations of HFD consumption (shown in Figure 3). All abbreviations are listed in the Abbreviations section.

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
