# Peer review of "Autophagy in High-Fat Diet and Streptozotocin-Induced Metabolic Cardiomyopathy: Mechanisms and Therapeutic Implications"

_ijms, 2025, doi:10.3390/ijms26041668_

Round 1

Reviewer 1 Report

Comments and Suggestions for Authors

This review works out the effect of a high fat diet and streptozotocin induced metabolic cardiomyopathy on autophagy. Overall, this is a well-researched and comprehensive review.

Some minor points of critique:

Line 89/90: Grammar problem. The authors write: In selective autophagy, isolation membrane formation is initiated around specific targets, facilitating their degradation [13].” Based on the referenced original literature I think the authors meant the generation of a membrane around a target, basically the formation of autophagosomes.

This review uses frequently terms like dysregulation, dysfunctional excessive, abnormal and others without specifying what this means. These are all non-specific and non-scientific terms which include changes of something that can be too much or too little. If possible, please avoid and used specific terms instead.

This review uses often abbreviated names of genes without introduction, which makes understanding at times difficult for those readers who are not familiar with the multiple layers that regulate autophagy. The authors have already included a table listing most -but not all- of the used abbreviations (for example ULK, FBXO27, RIPK are missing). Please revise the manuscript to include all abbreviations in this table.  

Line 77/79: Obesity increases the risk of developing T2DM, but the incidence rate of T2DM is also increasing among the normal weight and moderately overweight (BMI < 28) population as it ages.

Line 234: typo “pathologicaical” instead of pathological

Author Response

Journal: IJMS

Manuscript ID: ijms-3406826

Type: Review

Title: Autophagy in High-Fat Diet and Streptozotocin-Induced Metabolic Cardiomyopathy: Mechanisms and Therapeutic Implications

Authors: Rong Zhou , … , Xuehong Xu *

Dear Reviewer and Editor-in-chief,

According to the Comments and Suggestions from all three reviewers, we have made item-by-item responses as below, and made every modification in our revised manuscript in Red within our revision. We really appreciate the times and efforts from you and the reviewers!

The best regards,

Xuehong Xu, MS./PhD.  

Reviewer #1

Main comment: “This review works out the effect of a high fat diet and streptozotocin induced metabolic cardiomyopathy on autophagy. Overall, this is a well-researched and comprehensive review.”

Author response to reviewer main comment: We thank the reviewer for this comment on the importance of and our approach to this topic described in our manuscript.

Reviewer comment 1: “Line 89/90: Grammar problem. The authors write: In selective autophagy, isolation membrane formation is initiated around specific targets, facilitating their degradation [13].” Based on the referenced original literature I think the authors meant the generation of a membrane around a target, basically the formation of autophagosomes.”

Author response to Reviewer comment 1: We thank the reviewer for this comment and specific suggestion based on her/his critical reading. We had changed “In selective autophagy, isolation membrane formation is initiated around specific targets, facilitating their degradation [13]”is changed to “In selective autophagy, isolated membrane assembling around specific targets surrounds these targets, and constructs basically the formation of autophagosomes, which are facilitated their degradation [13]” (Page 3, Lines 89-91).

Reviewer comment 2: “This review uses frequently terms like dysregulation, dysfunctional excessive, abnormal and others without specifying what this means. These are all non-specific and non-scientific terms which include changes of something that can be too much or too little. If possible, please avoid and used specific terms instead.”

Author response to Reviewer comment 2: We thank the reviewer for this comment and specific suggestion. We had critical read and re-arranged the use of dysregulation thoroughly and comprehensively (Page 3, Lines 5, 25, 41; Page 4, Line 4; Page 9, Lines 32, 39-40; Page 15, Line 31).

Reviewer comment 3: “This review uses often abbreviated names of genes without introduction, which makes understanding at times difficult for those readers who are not familiar with the multiple layers that regulate autophagy. The authors have already included a table listing most -but not all- of the used abbreviations (for example ULK, FBXO27, RIPK are missing). Please revise the manuscript to include all abbreviations in this table.”

Author response to Reviewer comment 4: We thank the reviewer for this comment and specific suggestion. (1) We have inserted full term of ULK, FBXO27 and RIPK into their corrected locations (Page 5, Line 2; Page 8, Line 46; Page 6, Lines 14-15).

(2) We have checked entire manuscript and inserted all full term after the briefs in the revision.

(3) We have included all full term into Section “Abbreviations” (Page 19-20).

(4) We have also included all briefs in Table-1 (Page 5) and noted “All abbreviations are listed in section Abbreviations” underneath the table (Page 5, Line 4).

Reviewer comment 4: “Line 77/79: Obesity increases the risk of developing T2DM, but the incidence rate of T2DM is also increasing among the normal weight and moderately overweight (BMI < 28) population as it ages.”

Author response to Reviewer comment 5: We thank the reviewer for this comment and specific suggestion. We have made a change among the lines that the reviewer pointed out (Original, Line 77/79). We revised as “Additionally, obesity plays a critical role in the development of T2DM, and the progression of T2DM is typically accompanied by obesity” to “Additionally, obesity always possesses bad condition within the development of T2DM, therefore the progression of T2DM is typically accompanied by obesity” (Page 3, Lines 20-22).

Reviewer comment 5: “Line 234: typo “pathologicaical” instead of pathological”

Author response to Reviewer comment 2: We thank the reviewer for his critical reading. We had corrected typo “pathologicaical” and changed to “pathological” (Page 8, Line 5).

Reviewer 2 Report

Comments and Suggestions for Authors

The manuscript examines mitophagy, ferritinophagy, and ER-phagy, both selective and non-selective. This level of depth illuminates metabolic cardiomyopathy's molecular basis.

This work contributes to metabolic cardiomyopathy research and should be published.

The authors show their knowledge of current research. The manuscript is well-supported by scientific evidence.

The review emphasizes rodent models. Lack of human studies translation is a major issue.

Author Response

Journal: IJMS

Manuscript ID: ijms-3406826

Type: Review

Title: Autophagy in High-Fat Diet and Streptozotocin-Induced Metabolic Cardiomyopathy: Mechanisms and Therapeutic Implications

Authors: Rong Zhou , … , Xuehong Xu *

Dear Reviewer and Editor-in-chief,

According to the Comments and Suggestions from all three reviewers, we have made item-by-item responses as below, and made every modification in our revised manuscript in Red within our revision. We really appreciate the times and efforts from you and the reviewers!

The best regards,

Xuehong Xu, MS./PhD.  

Reviewer #2

Main comment: “The manuscript examines mitophagy, ferritinophagy, and ER-phagy, both selective and non-selective. This level of depth illuminates metabolic cardiomyopathy's molecular basis.”

Author response to reviewer main comment: We thank the reviewer for this comment on the importance of and our approach to this topic described in our manuscript.

Reviewer comment 1: “This work contributes to metabolic cardiomyopathy research and should be published.”

Author response to Reviewer comment 1: We thank the reviewer for this comment on the importance of and his critical reading on our entire manuscript, which constructively help us to accomplish this revised manuscript.

Reviewer comment 2: “The authors show their knowledge of current research. The manuscript is well-supported by scientific evidence.”

Author response to Reviewer comment 2: We thank the reviewer for this comment. We had follow all suggestions from this reviewer and other reviewer to eventually accomplish this “better-looking” revision. We had made all changes that reviewers’ Comments and Suggestions, which we highlighted in Red within this revision. All changes were whereabouts labeling with Page-Line format.

Reviewer comment 3: “The review emphasizes rodent models. Lack of human studies translation is a major issue.”

Author response to Reviewer comment 3: We thank the reviewer for this comment and specific suggestion. We had made some changes by providing extra information of human issues. We added an extra Sub-section,  in Red, entitled “5.4. Caloric Restriction and Physical Exertion Prevent Metabolic Cardiomyopathy in Human” (Page 17, Lines 26-53 to Page 18, Lines 1-26), along with an extra figure (Figure 4). We also added “autophagy and human health”, in Red, in section “6. Conclusion and perspective” (Page 19, Lines 21-33).

Reviewer 3 Report

Comments and Suggestions for Authors

Zhou et al. summarize the mechanisms and therapeutic implications of high-fat diet- and STZ-induced cardiac autophagy in their comprehensive review. The topic is interesting and has significant clinical importance. The text is well-structured and easy to read despite the detailed description of the molecular pathways.  

1. Table 1: please change "Vitro" to "in vitro" in the second (models) column.

2. LT-HFD is not introduced in the text and the figure legend of Fig. 1.

3. It would be helpful to show a summary figure or table on the therapeutic interventions reviewed in Section 5.

4. Is there any clinical study investigating the effects of the natural extracts or pharmaceutical agents reviewed in section 5 on metabolic cardiomyopathy? Please discuss whether they had beneficial effects on metabolic cardiomyopathy.  

Author Response

Journal: IJMS

Manuscript ID: ijms-3406826

Type: Review

Title: Autophagy in High-Fat Diet and Streptozotocin-Induced Metabolic Cardiomyopathy: Mechanisms and Therapeutic Implications

Authors: Rong Zhou , … , Xuehong Xu *

Dear Reviewer and Editor-in-chief,

According to the Comments and Suggestions from all three reviewers, we have made item-by-item responses as below, and made every modification in our revised manuscript in Red within our revision. We really appreciate the times and efforts from you and the reviewers!

The best regards,

Xuehong Xu, MS./PhD.  

Reviewer #3

Main comment: “Zhou et al. summarize the mechanisms and therapeutic implications of high-fat diet- and STZ-induced cardiac autophagy in their comprehensive review. The topic is interesting and has significant clinical importance. The text is well-structured and easy to read despite the detailed description of the molecular pathways.”

Author response to reviewer main comment: We appreciate the critical reading, and constructive Comments and Suggestions (C-and-S). We have followed the C-and-S and made all changes/corrections to produce this version.

Reviewer comment 1: “1. Table 1: please change "Vitro" to "in vitro" in the second (models) column.”

Author response to Reviewer comment 1: We thank the reviewer for this specific suggestion. We have changed “Vitro”in Table 1 to "in vitro" in Red within this revision (Page 5).

Reviewer comment 2: “2. LT-HFD is not introduced in the text and the figure legend of Fig. 1.”

Author response to Reviewer comment 2: We thank the reviewer for this comment.

(1) We have added “Long term high fat diet” in the figure legend of Figure. 1 by changing “LT-HFD” to “Long term high fat diet, LT-HFD” (Page 6, Line 196), which was highlighted with Red.

(2) We have inserted “Long term high fat diet” in main body (Page 11, Lines 390) changing “LT-HFD” to “long term high fat diet (LT-HFD)”, which was highlighted with Red and where first appeared in the main text.

(3) we have checked entire manuscript and added full name to every abbreviated terms including SQSTM1/p62, TRIM13, UFL1, C53 and Atg8 (Page 3, Paragraph 3); LC3II, Rab9, Pink1, Park2, Mfn1, Rab7, and Parkin (Page 4, Paragraphs 2 and 5); FOXO1  (Page 7, Figure legend); LAMP2 and FBXO27 (Page 9, Paragraph 1); FAM134B, SEC62, RTN3 , CCPG1, ATL3, TEX264 (Page 14, Paragraph 2); eIF2α, GRP78, ATF4, PERK, IRE1α, ATF6α and CHOP (Page 15, Paragraph 2; Page 16, Paragraph 1).

Reviewer comment 3: “3. It would be helpful to show a summary figure or table on the therapeutic interventions reviewed in Section 5.”

Author response to Reviewer comment 3: We thank the reviewer for this comment and constructive suggestion. We had made some changes by providing extra information of human issues by adding an extra figure (Figure 4). We also added “autophagy and human health”, in Red, in section “6. Conclusion and perspective” (Page 19, Lines 21-33).

Reviewer comment 4: “4. Is there any clinical study investigating the effects of the natural extracts or pharmaceutical agents reviewed in section 5 on metabolic cardiomyopathy? Please discuss whether they had beneficial effects on metabolic cardiomyopathy.”

Author response to Reviewer comment 4: We thank the reviewer for this comment and specific suggestion. According to reviewers’ Comments and Suggestions, we added an extra Sub-section, in Red, entitled “5.4. Caloric Restriction and Physical Exertion Prevent Metabolic Cardiomyopathy in Human” (Page 17, Lines 26-53 to Page 18, Lines 1-26), along with an extra figure (Figure 4). We also added “autophagy and human health”, in Red, in section “6. Conclusion and perspective” (Page 19, Lines 21-33).